# Benchmarking a well-calibrated measure of weight similarity of deep neural network models

## Abstract

Deep learning approaches have revolutionized artificial intelligence, but model opacity and fragility remain significant challenges. The reason for these challenges, we believe, is a knowledge gap at the heart of the field — the lack of well-calibrated metrics quantifying the similarity of the internal representations of models obtained using different architectures, training strategies, different checkpoints, or under different random initializations. While several metrics have been proposed, they are poorly calibrated and susceptible to manipulations and confounding factors, as well as being computationally intensive when probed with a large and diverse set of test samples. We report here an integration of chain normalization of weights and centered kernel alignment that, by focusing on weight similarity instead of activation similarity, overcomes most of the limitations of existing metrics. Our approach is sample-agnostic, symmetric in weight space, computationally efficient, and well-calibrated.

## 1 Introduction

In the last few decades, deep learning has revolutionized the study and development of artificial intelligence (AI). The revolution has been driven by the almost dizzying introduction of novel model architectures: fully connected perceptrons, deep convolutional neural networks (Krizhevsky et al., 2012), transformers (Dosovitskiy et al., 2020), diffusion models (Ho et al., 2020) and, most recently, large language models. These approaches have all achieved remarkable success, pushing the state of the art across numerous fields, including computer vision, natural language processing, and speech recognition.

In contrast to prior approaches in computer science, where developing parsimonious models using limited resources was the goal, the path pursued in AI has been one in which increased performance has been achieved at the cost of gigantic models using vast amounts of resources. For example, a modern large language model can easily contain billions of parameters and requires massive computational and data resources to train, resources that are only available to the largest organizations.

On par with resource requirements is the fact that increasing model complexity comes at the cost of model opacity. The exact nature of what a perceptron model learns *still* remains elusive (Calude et al., 2023), but the lack of understanding grows exponentially larger as more complex architectures are considered (Rudin, 2019).

This lack of interpretability of model predictions is made all the more worrisome because many neural network architectures are known to be vulnerable to adversarial attacks (Goodfellow et al., 2014), exhibit generalization gaps, and are prone to shortcut learning (DeGrave et al., 2021; Geirhos et al., 2020). All these concerns dramatically limit the ability to deploy deep learning models to high-stakes settings, such as healthcare systems (T. Dhar et al., 2023).

In view of these concerns, it is not surprising that multiple lines of research are dedicated to improving the interpretability, generalizability, and robustness of deep learning models. One fruitful approach has been the development of attribution methods Simonyan et al. (2013); Sundararajan et al. (2017); Selvaraju et al. (2019) that aim to identify the specific features driving model predictions.

However, recent studies have revealed a lack of consensus and, at times, inaccurate attributions, raising concerns about the fidelity of these methods (Saporta et al.; Rudin, 2019).

Another fruitful approach is the investigation of how specific training manipulations can address known concerns — model stitching (Bansal et al., 2021), component ablation experiments (Shah et al.), data augmentation such as adversarial training (Goodfellow et al., 2014), early stopping (Pang et al., 2020), and batch normalization (Ioffe & Szegedy, 2015) have all been used in an attempt to increase model robustness. Despite their widespread utilization, these techniques still rely on "trial and error," "common practices," and "rules of thumb," rather than systematized knowledge.

It is our contention that the limitations of all the approaches discussed above are due to a gap at the heart of the field — the lack of a trustworthy, well-calibrated measure of model similarity. To address the limitations of current approaches, we need to quantify whether different neural networks learn in a similar or distinct manner at various stages of internal processing (Klabunde et al.) or whether there are universal or idiosyncratic mechanisms underlying reported high performance.

Li et al. (2015) introduced the concept of convergent learning, which asks whether different neural networks — separately trained with varying random initializations, different architectures, disjoint data samples, or optimization algorithms — ultimately learn the same underlying representations. Various research groups have proposed representational similarity metrics aiming to quantify the (dis)similarity between *activation values* computed by different layers or models for a given set of inputs. The most widely studied representational similarity metrics are canonical correlation analysis (CCA) (Morcos et al.), Procrustes (Gower & Dijksterhuis, 2004), and centered kernel alignment (CKA) (Kornblith et al., 2019).

Despite the valuable insights provided by research on representational similarity in interpreting deep neural networks, there remains a lack of consensus on interpreting the outputs of different representational similarity metrics (Ding et al., 2021; Cui et al., 2024). For example, Davari et al. (2023) characterized in detail the sensitivity of CKA to data transformations that do not lead to functional changes for neural networks, demonstrating that CKA can be easily manipulated. The dependence of CKA on specific input manipulations is to be expected, given the nature of representation activation values calculated based on a subset of probing inputs (Fig. 1, Fig. 5).

Here, we propose a new approach. We start from the observation that learned knowledge in a neural network is captured in the values of the weights acquired during training. Despite their fundamental importance, there is surprisingly little research on the weight similarity of neural networks. Wang et al. (2022) made the first attempt to explore weight similarity by proposing a chained normalization operator that ensures invariance of weight similarity to permutation of neurons, i.e., the shuffling of neurons within the same layer. Inspired by their work, as well as the foundations of CKA, we propose a novel weight similarity metric that applies CKA to a kernel of chained, normalized weights. Our approach — which we denote weights CKA (wCKA) — shifts the focus from activation-based comparisons to weight-based comparisons. This is an important shift because weights capture the learned parameters of a model, independent of specific inputs and invariant to input perturbations. This idea offers more stable and generalizable insights about model similarity and the ability to deal with input-dependent and spurious similarities.

Our main contributions can be summarized as follows:

- We propose a novel metric quantifying the similarity of neural networks in terms of learned parameters — model weights. The proposed metric is invariant to permutation and intertwiner (Godfrey et al., 2022) transformations, independent of probing input, and computationally efficient (Fig. 1c).

- We benchmark wCKA against three existing representational similarity metrics — Procrustes, CKA, and dCKA — with random initialized neural networks and similar neural networks obtained from successive training epochs. In contrast to other metrics, wCKA demonstrates robust differentiation power in the calibration task towards the number of probing samples and out-of-distribution corruptions,

- We validate the reliability of wCKA similarity estimates through the analysis of their correlation with functional similarity, quantified by the fraction of agreed predictions on a variety of test samples, including out-of-distribution corruptions and adversarial attacks.

## 2 METHODS

### 2.1 REPRESENTATION SIMILARITY

Representational similarity metrics measure the similarity between representations, i.e., activation values of different layers or models in response to a given set of inputs. Let $\mathbf{X}_1 \in \mathbb{R}^{n \times d_1}$ and $\mathbf{X}_2 \in \mathbb{R}^{n \times d_2}$ denote the activation matrices of two different layers or models, where $n$ is the number of input samples, e.g. images, and $d_1$ and $d_2$ are the dimensionalities of the activations, i.e. number of neurons at correspondent layers. Each row of $\mathbf{X}_1$ or $\mathbf{X}_2$ corresponds to the activation pattern of the layer for a specific input example, and each column corresponds to the activation values of $n$ input samples for a single neuron.

#### 2.1.1 PROCRUSTES

Procrustes measure the similarity between two representation matrices by minimizing the Frobenius norm of the difference between the two matrices. The Procrustes distance is defined as:

$$d_{\text{Procrustes}}(\mathbf{X}_1, \mathbf{X}_2) = \|\mathbf{X}_1\|_F^2 + \|\mathbf{X}_2\|_F^2 - 2\|\mathbf{X}_1^\top \mathbf{X}_2\|_*$$

where $\| \cdot \|_F$ denotes the Frobenius norm (Szabo, 2015), and $\| \cdot \|_*$ denotes the nuclear norm (Manngård et al., 2017).

#### 2.1.2 CKA

Kornblith et al. (2019) proposed centered kernel alignment (CKA) as a way to link the representational similarity of two models to the inner product of features. They argued that CKA exhibits desired invariance properties: invariant to orthogonal transformation and isotopic scaling. CKA is defined as:

$$\text{C}(\mathbf{K}_1, \mathbf{K}_2) = \frac{\text{S}(\mathbf{K}_1, \mathbf{K}_2)}{\sqrt{\text{S}(\mathbf{K}_1, \mathbf{K}_1) \cdot \text{S}(\mathbf{K}_2, \mathbf{K}_2)}} \tag{1}$$

where $\mathbf{K}_1$ and $\mathbf{K}_2$ are kernel matrices of $\mathbf{X}_1$ and $\mathbf{X}_2$, respectively:

$$\mathbf{K}_1^{ij} = k_1(\mathbf{x}_{1i}, \mathbf{x}_{1j}) \quad \text{and} \quad \mathbf{K}_2^{ij} = k_2(\mathbf{x}_{2i}, \mathbf{x}_{2j}),$$

where $k_1$ and $k_2$ are kernel functions, and $\mathbf{x}_{1i}$ and $\mathbf{x}_{2i}$ are rows (i.e., activation vector of a specific input sample) of $\mathbf{X}_1$ and $\mathbf{X}_2$, respectively, and $\text{S}(\mathbf{K}_1, \mathbf{K}_2)$ is the Hilbert-Schmidt Independence Criterion (Gretton et al., 2005) between $\mathbf{K}_1$ and $\mathbf{K}_2$, computed as:

$$\text{S}(\mathbf{K}_1, \mathbf{K}_2) = \frac{1}{(n-1)^2} \text{tr}(\mathbf{K}_1 \mathbf{H} \mathbf{K}_2 \mathbf{H}) \tag{2}$$

where $\mathbf{H} = \mathbf{I}_n - \frac{1}{n}\mathbf{1}_n\mathbf{1}_n^\top$ is the centering matrix.

Most researchers using CKA use a linear kernel. In this case, CKA reduces to:

$$\text{C}_{\text{linear}}(\mathbf{X}_1, \mathbf{X}_2) = \frac{\|\mathbf{X}_2^\top \mathbf{X}_1\|_F^2}{\|\mathbf{X}_1^\top \mathbf{X}_1\|_F \cdot \|\mathbf{X}_2^\top \mathbf{X}_2\|_F} \tag{3}$$

where $\| \cdot \|_F$ again denotes the Frobenius norm.

Nguyen et al. (2020) utilized minibatch CKA, which approximates the original CKA by averaging results from batches instead of the entire data population, thereby reducing memory cost, to reveal different internal representations learned by wide vs. deep neural networks. Further, Godfrey et al. (2022) proposed variants of Procrustes and CKA that are invariant to model symmetry groups — intertwiner groups — for neural networks with $ReLu$ activation function.

#### 2.1.3 DECONFOUNDED CKA

Cui et al. (2022) proposed de-confounded CKA (dCKA) to regress out the spurious similarity in CKA due to similar data structure in the probing samples. The dCKA is defined as:

$$\text{C}_d(\mathbf{K}_1, \mathbf{K}_2) = \text{C}(\tilde{\mathbf{K}}_{d1}, \tilde{\mathbf{K}}_{d2})$$

where $\tilde{\mathbf{K}}_{d1}, \tilde{\mathbf{K}}_{d2}$ are positive semidefinite approximations of deconfounded kernel matrices $\mathbf{K}_{d1}, \mathbf{K}_{d2}$, respectively, by removing negative eigenvalues, where

$$\mathbf{K}_{d1} = \mathbf{K}_1 - \hat{\alpha}_1 \mathbf{K_0}$$

$$\mathbf{K}_{d2} = \mathbf{K}_2 - \hat{\alpha}_2 \mathbf{K_0}$$

where $\hat{\alpha}_1$ and $\hat{\alpha}_2$ are regression coefficient minimizing the Frobenius norm of $\mathbf{K}_{d1}$ and $\mathbf{K}_{d2}$, and $\mathbf{K_0}$ represents similarity of input data structure $\mathbf{K_0} = l(\mathbf{X}, \mathbf{X})$ and $l$ represents the same kernel function as in CKA. Assuming linear and additive confounding effect, $\hat{\alpha}_1$ and $\hat{\alpha}_2$ can be computed as:

$$\hat{\alpha}_1 = \left(\text{vec}(\mathbf{K_0})^\top \text{vec}(\mathbf{K_0})\right)^{-1} \text{vec}(\mathbf{K_0})^\top \text{vec}(\mathbf{K}_1)$$

$$\hat{\alpha}_2 = \left(\text{vec}(\mathbf{K_0})^\top \text{vec}(\mathbf{K_0})\right)^{-1} \text{vec}(\mathbf{K_0})^\top \text{vec}(\mathbf{K}_2)$$

where $\text{vec}(\mathbf{K})$ represents the vectorization of matrix $\mathbf{K}$.

## 2.2 Weights Centered Kernel Alignment (wCKA)

Instead of comparing activations, our proposed wCKA operates on the weight matrices of neural networks. It builds on Wang et al. (2022), which proposed a weight normalization operator that is invariant to re-parameterization, such as the shuffling of neurons within the same layer. Wang et al. (2022) weight normalization operator $\phi$ is defined as:

$$\phi(W_1, W_2, \ldots, W_l) = W_1 W_2 \ldots W_\ell W_\ell^\top \ldots W_2^\top W_1^\top \tag{4}$$

where $W_1, W_2, \ldots, W_\ell$ are the weight matrices of a neural network with $\ell$ layers.

Now, let $W_1^{(i)} \in \mathbb{R}^{d_1^{i-1} \times d_1^l}$ and $W_2^{(i)} \in \mathbb{R}^{d_2^{i-1} \times d_2^l}$ represent the weight matrices of two neural networks for layer $i$, where $d_1^{i-1}$ and $d_2^{i-1}$ are the number of neurons in the previous layers, and $d_1^l$ and $d_2^l$ are the number of neurons in the current layers, respectively, and define the kernels

$$\mathbf{K}_1 = \phi(W_1^{(1)}, W_1^{(2)}, \ldots, W_1^{(l)}) \quad \text{and} \quad \mathbf{K}_2 = \phi(W_2^{(1)}, W_2^{(2)}, \ldots, W_2^{(l)}).$$

Plugging these kernels into Eq. (1) yields

$$C_w(\mathbf{W_1}, \mathbf{W_2}) = \frac{\|\mathbf{W}_2^\top \mathbf{W}_1\|_F^2}{\|\mathbf{W}_1^\top \mathbf{W}_1\|_F \cdot \|\mathbf{W}_2^\top \mathbf{W}_2\|_F} \tag{5}$$

where

$$\mathbf{W}_1 = W_1^{(1)} W_1^{(2)} \ldots W_1^{(l)} \quad \text{and} \quad \mathbf{W}_2 = W_2^{(1)} W_2^{(2)} \ldots W_2^{(l)}$$

## 2.3 Invariance to permutation and intertwiner transformations

Wang et al. (2022) proved invariance of the chain normalization operator to permutation, and CKA is also known to be invariant to orthogonal transformation (Kornblith et al., 2019). These properties ensure the invariance of wCKA to permutation.

Godfrey et al. (2022) introduced the concept of intertwiner groups, which are groups of transformations that modify the model weights while preserving the underlying function of the neural network. Let $W := \{W^{(i)} \mid i = 1, \ldots, k\}$ be the collection of all weights of a $k$-layer fully connected neural network. According to Proposition 3.4 in Godfrey et al. (2022), weights $\mathbf{W}$ and $\mathbf{W}'$ are functionally equivalent under the following transformation:

$$\mathbf{W}' = (W^{(1)} A_1, \phi(A_1^{-1}) W^{(2)} A_2, \ldots, \phi(A_{k-1}^{-1}) W^{(k)})$$

where $A_i \in G_{\sigma_i}$, the intertwiner group defined for the activation function $\sigma_i$:

$$G_{\sigma_i} := \{A \in GL_{n_i}(\mathbb{R}) \mid \exists B \in GL_{n_i}(\mathbb{R}) \text{ such that } \sigma_i \circ A = B \circ \sigma_i\}$$

where $GL_{n_i}(\mathbb{R})$ represents the general linear group of invertible matrices in $\mathbb{R}^{n_i \times n_i}$, and $\phi$ is defined as:

$$\phi_\sigma(A) = \sigma(A)\sigma(I_n)^{-1}$$

where $I_n$ is the identity matrix of size $n$. Please note that our notation for weight matrices $W_i$ differs from that in Godfrey et al. (2022), as our rows and columns are transposed. Specifically, we use $\sigma(xW_i)$, whereas they use $\sigma(W_i^T x + b)$ as the layer function.

Godfrey et al. (2022) show that $\phi_\sigma(A) = A$ for four types of activation functions: $\sigma(x) = x$ (identity), $\sigma(x) = \frac{e^x}{1+e^x}$ (sigmoid), $\sigma(x) = \text{ReLU}(x)$, and $\sigma(x) = \text{LeakyReLU}(x)$. We show here that wCKA is invariant to the intertwiner transformation described above for these four types of activation functions:

$$\mathbf{W'_1} = W_1^{(1)} A_1 \, \phi(A_1^{-1}) W_1^{(2)} A_2 \ldots \phi(A_{k-1}^{-1}) W_1^{(k)}$$

Since $\phi_\sigma(A_i^{-1}) = A_i^{-1}$, we have:

$$\mathbf{W'_1} = W_1^{(1)} A_1 \, A_1^{-1} W_1^{(2)} A_2 \ldots A_{k-1}^{-1} W_1^{(k)}$$

terms $\phi_\sigma(A_i^{-1}) A_i = I$, thus can be canceled out, therefore:

$$\mathbf{W'_1} = W_1^{(1)} W_1^{(2)} \ldots W_1^{(k)} = \mathbf{W_1}$$

Similarly,

$$\mathbf{W'_2} = W_2^{(1)} W_2^{(2)} \ldots W_2^{(k)} = \mathbf{W_2}$$

Thus,

$$\mathbf{C}_w(\mathbf{W'_1}, \mathbf{W'_2}) = \mathbf{C}_w(\mathbf{W_1}, \mathbf{W_2})$$

This proves the invariance of wCKA under the intertwiner group transformation for neural networks with identity, sigmoid, ReLU, or LeakyReLU activation functions.

### 2.4 FUNCTIONAL SIMILARITY

In addition to internal representations and weights, neural networks may be dissimilar in their functionality as well. The intrinsic connection between representational and functional dissimilarity is well recognized (Klabunde et al.) and serves as a fundamental rationale for benchmarking representational similarity metrics in Ding et al. (2021).

To facilitate a standard and systematic evaluation of representational similarity metrics, Ding et al. (2021) introduced a benchmarking framework that emphasizes the intrinsic connection between representational and functional similarity. Essentially, the idea is that if two neural network models exhibit different performances on certain tasks, they must learn different internal representations. Therefore, similarity metrics can be evaluated by the rank correlation between metric distance and functional performance.

To further benchmark wCKA against other similarity metrics, we adopt and extend this benchmarking framework. Instead of quantifying the functional distance of two models by their accuracy gap, *we use the fraction of agreed predictions between two models* across clean test samples, out-of-distribution corruptions, and adversarially attacked samples:

$$S(O, O') = \frac{1}{N} \sum_{i=1}^{N} 1 \left\{ \arg\max_j O_{i,j} = \arg\max_j O'_{i,j} \right\}.$$

where $O$ is a vector of model predictions, and $i$ and $j$ are indices capturing the evaluated samples and class labels, respectively.

### 2.5 STATISTICAL TESTING

We measure the correlation between wCKA and functional similarity using both Pearson's linear correlation coefficient $r$ and Spearman's rank correlation $\rho$. We then Fisher transform the correlation coefficient to an approximate $z$-score:

$$z = \text{atanh}(r)$$

with standard error $\frac{1}{\sqrt{n-3}}$, where $n$ is the sample size. We perform statistical testing and $p$-value calculation on $z$-scores and back transform to $r$ to obtain confidence intervals.

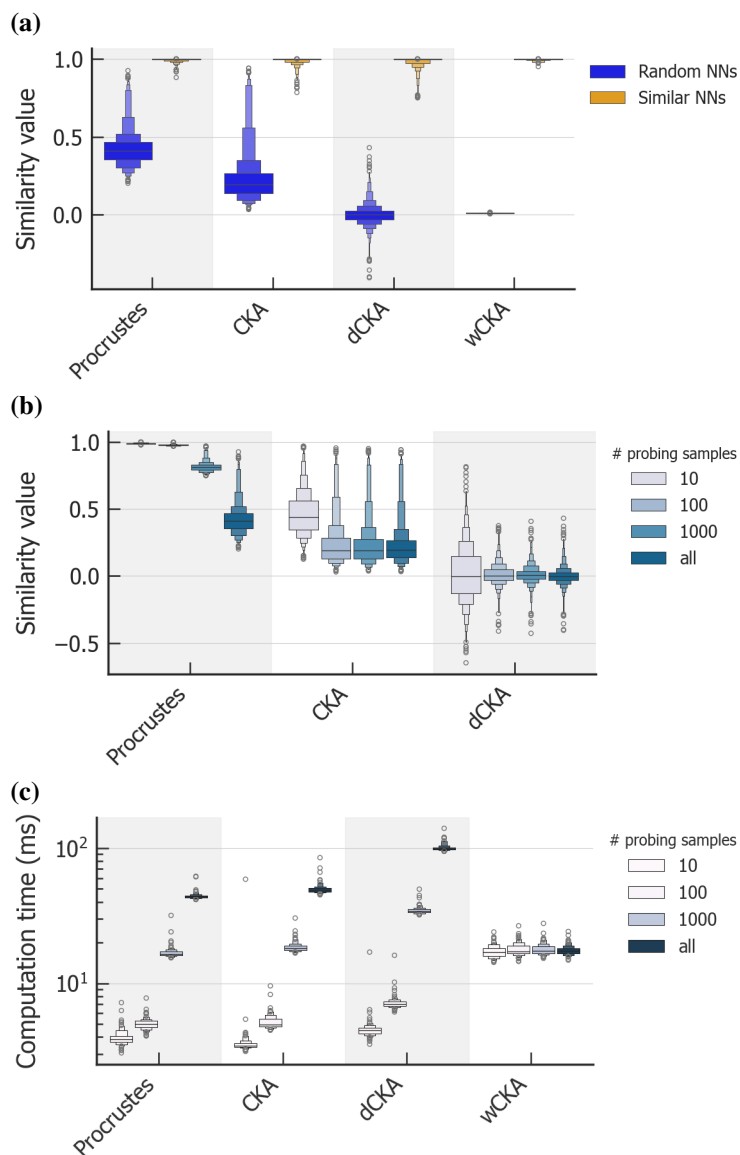

Figure 1: **Calibration of neural network model similarity metrics and computational efficiency.**
**(a)** Model similarity for different model similarity metrics for neural networks (NN) with randomly
selected weights (dark blue) and weights of a single model at consecutive training epochs (orange).
We use enhanced box plots to capture the distribution of observed similarity of activations across
3,417 test images for Procrustes, CKA, and dCKA. One would expect similarity to be close to zero
for random NN models and close to 1 for very similar models. Instead, we observe overlapping
distributions of similarity values for two of the three models. For comparison, we show that the
calibration of wCKS is nearly perfect. **(b)** Impact of a number of probing images on the activation
similarity values of the three metrics for random NN. It is visually apparent that calibration improves
dramatically as the number of test images increases from 10 to 1,000, but then it saturates. Note
that we do not plot data for wCKA as it does not use probing images to estimate similarity. **(c)**
Computational time for different metrics across varying numbers of probing samples. Enhanced box
plots illustrate the distribution of computational time taken for computing each of the four metrics
on fully connected NNs with four hidden layers of thirty-two neurons in each layer, based on 100
runs. The computational time of Procrustes, CKA, and dCKA scale up with an increasing number
of probing samples, while that of wCKA remains consistently low.

## 3 RESULTS

### 3.1 wCKA METRIC HAS SUPERIOR CALIBRATION TO CURRENT METRICS

We first characterize the calibration characteristics of the four metrics considered here: Procrustes, CKA, dCKA, and wCKA metrics. Specifically, we create ensembles of neural network models for which we expect similarity to be close to zero or close to 1. Starting from fully connected neural networks with four hidden layers, each with 32 neurons, which we train handwritten digits of zero, one, and two from the MNIST dataset. We store model checkpoints from adjacent epochs after 95 epochs of training when the performance of models has already converged. We set 100 randomly initialized pre-trained networks as the ensemble of random neural networks with expected zero similarity. We set 100 models from adjacent training epochs after convergence as the ensemble of similar neural networks with an expected similarity of 1. We compute activation similarity for Procrustes, CKA, and dCKA on the 3,147 test images — the entire test set — probing samples randomly selected from clean or 15 out-of-distribution corruptions. We compute wCKA directly on chain-normalized weight matrices.

As was pointed out by Davari et al. (2023); Cui et al. (2022), the Procrustes and CKA display spurious similarities for random neural networks, indicating significant estimation bias (Fig. 1a). Even dCKA, which yields unbiased similarity values, displays a large estimation uncertainty with some values of activation similarity greater than 0.3. as high as 0.5 when it should be zero as some values smaller than 0.8 when it should be close to 1. In contrast, wCKA consistently yields values close to zero for random neural networks and close to one for similar neural networks (Fig. 1a).

However, calibration is not the only concern with Procrustes, CKA, and dCKA. All three approaches quantify activation similarity. That is, they must be probed with test images. Figure 1b shows that estimates of similarity for Procrustes and CKA converge slowly with a number of probing images to their biased estimates. Similarly, dCKA displays slow converging estimation uncertainty.

### 3.2 wCKA MODEL SIMILARITY MORE ACCURATELY CAPTURES THE FUNCTIONAL SIMILARITY OF FULLY CONNECTED NEURAL NETWORKS

Next, we systematically evaluate the ability of these different measures of model similarity to capture the functional similarity of the model. Figure 2 illustrates our multifactorial benchmarking pipeline. We compare model similarity and functional similarity for models with different archi-

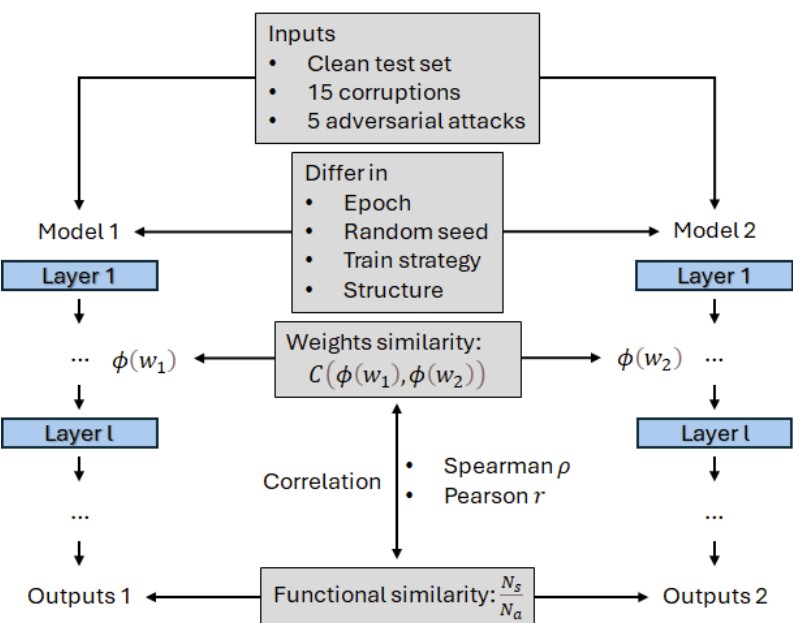

Figure 2: **Schematic illustration of our multifactorial benchmarking experimental pipeline.**

tectures, initialized using different random seeds, trained to different epochs, and using different training strategies. We consider five model architectures: fully connected neural networks with one hidden layer of eight neurons in each layer; fully connected neural networks with one hidden layer of thirty-two neurons in each layer; fully connected neural networks with four hidden layers of eight neurons each layer; fully connected neural networks with four hidden layers of thirty-two neurons each layer; and fully connected neural networks with ten hidden layer of thirty-two neurons each layer. We consider three different training strategies: standard training and two adversarial training approaches (FGSM (Goodfellow et al., 2014) and PGD (Madry et al., 2019)).

We capture the performance of a given model similarity by correlating the value for two models with the functional similarity, that is, the fraction of test images with identical predicted classification (Fig. 3). We probe the functional similarity of two neural networks predictions not only on clean test images but also on fifteen types of image corruption (MNIST-C dataset Mu & Gilmer (2019)) and five types of adversarial attack (FGSM (Goodfellow et al., 2014), Fast FGSM (Wong et al., 2020), Gaussian noise, PGD (Madry et al., 2019), and TRADES (Zhang et al., 2019)).

We calculate both Pearson's linear correlation coefficient $r$ as well as Spearman's rank correlation $\rho$ and apply Fisher's transformation to estimate confidence intervals and statistical significance. Figure 3 demonstrates that all metrics perform similarly well for recognizing that two different checkpoints (epochs) of the same model are quite similar. However, for all other conditions, wCKA displays stronger Pearson's linear correlation — indicating better calibration — as well as higher

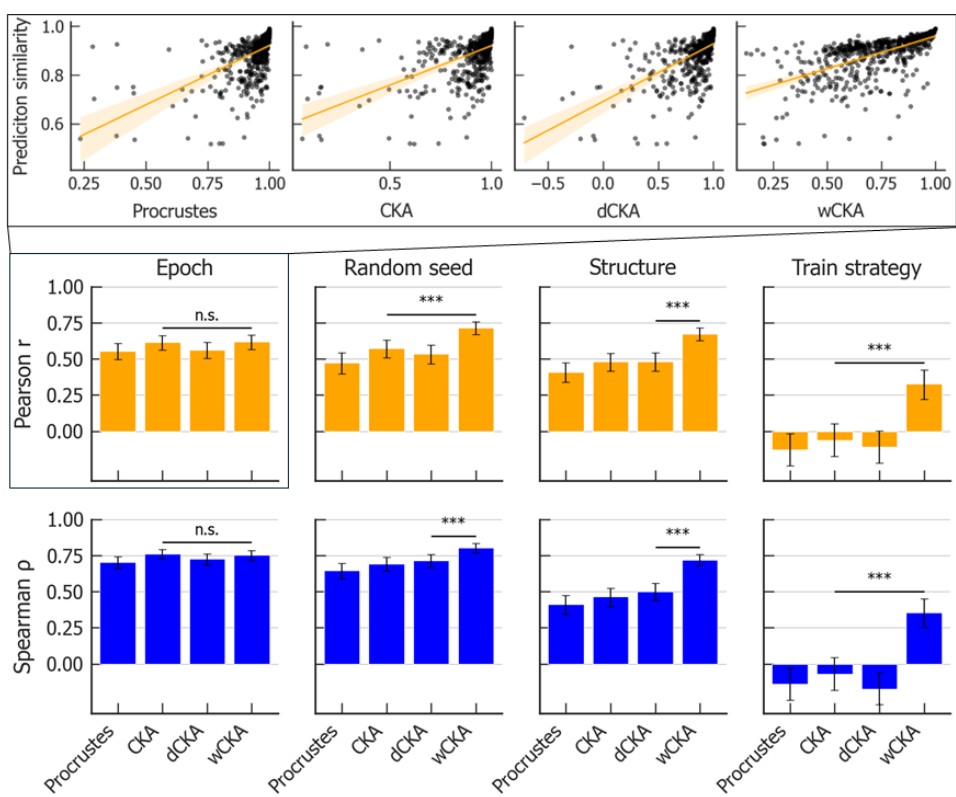

Figure 3: **Benchmarking of similarity metrics on trained-from-scratch, fully connected neural networks.** **(Top)** We calculate the correlation between activation or weight model similarity and functional similarity for two models differing in one of 4 possible ways. The difference in the data shown is the training epoch. **(Middle)** Pearson's $r$ and **(Bottom)** Spearman's $\rho$ estimated for pairs of models differing in the four ways denoted at the top of the column for the four metrics considered. Error bars show the 95% confidence intervals for the estimate of the mean. Horizontal lines connect correlation coefficient estimates being compared. We use "n.s." to indicate a lack of statistical significance of the difference and "***" to indicate statistical significance at the $p < 0.001$ level.

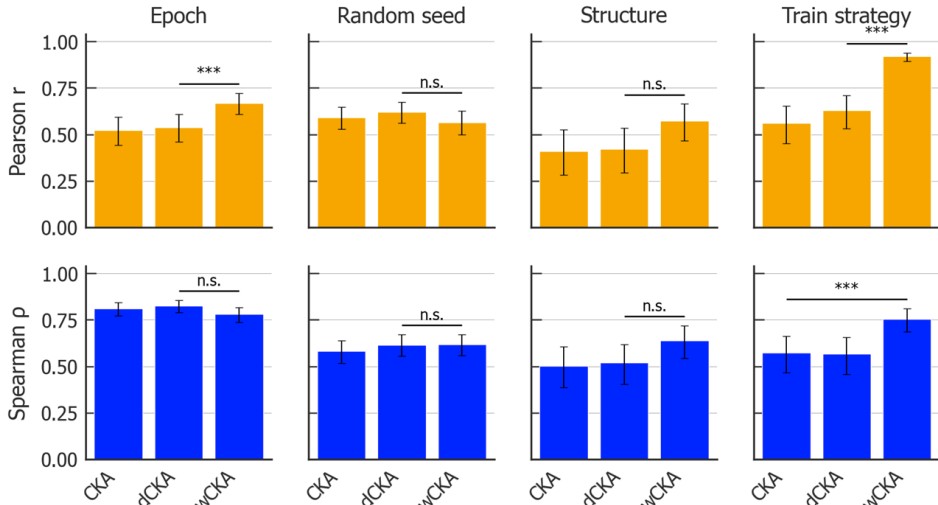

Figure 4: **Benchmarking of similarity metrics on fine-tuned, convolutional neural networks.** **(Top)** We calculate the correlation between activation or weight model similarity and functional similarity for two models differing in one of 4 possible ways. The difference in the data shown is the training epoch. **(Middle)** Pearson's $r$ and **(Bottom)** Spearman's $\rho$ estimated for pairs of models differing in the four ways denoted at the top of the column for the four metrics considered. Error bars show the 95% confidence intervals for the estimate of the mean. Horizontal lines connect correlation coefficient estimates being compared. We use "n.s." to indicate a lack of statistical significance of the difference and "***" to indicate statistical significance at the $p < 0.001$ level.

Spearman's rank correlation between metric value and functional similarity. Even for models trained using different strategies, wCKA is able to identify some degree of similarity between the models, something the other metrics fail to do.

### 3.3 WCKA MODEL SIMILARITY MORE ACCURATELY CAPTURES THE FUNCTIONAL SIMILARITY OF FINE-TUNED, CONVOLUTIONAL NEURAL NETWORKS

For the last decade, transfer learning has become immensely popular Rasmy et al.. The reason is that fine-tuning pre-trained deep learning models for new domain-specific tasks is a promising way to overcome resource limitations. In transfer learning, one typically fine-tunes a layer of perceptrons, i.e., fully connected neural networks. However, transfer learning, if anything, exacerbates the issue of lack of interpretability. Thus, it is also needed, in this context, to develop trustworthy methods for quantifying model similarity.

To test the applicability of wCKA to transfer learning, we train a simple convolutional neural network with two convolutional layers, with 64 and 32 kernels of size 5x5, respectively, and one fully connected hidden layer of 1,024 neurons, on the full dataset of MNIST containing ten classes of handwritten images. This illustrative network achieves its highest validation performance on the 24th epoch. We then freeze the convolutional layers and fine-tune the weights of specific fully connected hidden layers on a simpler task: classifying handwritten digits for zero, one, and two.

Following again the pipeline described in Fig. 2, we vary four characteristics of the trained models. We consider three distinct architectures for the fully connected layers: one hidden layer of 32 neurons, one hidden layer of 1024 neurons, and ten hidden layers of 32 neurons each. We fine-tune the training of the fully connected layer using either standard or PGD adversarial training. We consider different random initializations and different checkpoints. This procedure, though conducted on very simple model architectures and tasks, nonetheless closely resembles the typical transfer learning procedure (Ferreira et al.).

Figure. 4 shows the results of our benchmarking. It is noteworthy that the estimated similarities calculated by all metrics are not as negatively affected when comparing models trained using different training strategies. This is particularly striking for Procrustes, CKA, and dCKA, which, when con-

sidering full training, performed so poorly. The explanation is likely the fact that the convolutional layers are frozen, which imposes a higher degree of similarity even under adversarial training. It is nonetheless visually apparent that wCKA estimates model similarities that *also* consistently display higher correlation with functional similarity than prior metrics. As wCKA is agnostic to frozen original networks, it can be readily integrated with any type of large model and shed light on the fine-tuning process of fully connected layers — highlighting its robustness.

# 4 DISCUSSION

We addressed in this study a major knowledge gap at the heart of deep learning — the lack of a trustworthy, well-calibrated measure of model similarity. This gap has so far prevented the type of progress that one would hope for with regard to understanding how neural network models can be so opaque and fragile. Here, we present a well-calibrated metric that can capture model similarity under a number of critical perturbations: model architecture, training strategy, checkpoints, or random initialization. Our benchmarking experiments demonstrate that wCKA displays superior calibration characteristics: matching setpoints and linearity of relationship to functional similarity.

Our approach — to the best of our knowledge — pioneers the integration of chain normalization of weight matrices with centered kernel alignment, a widely used similarity metric for estimating similarity between the internal representations of two neural network models. wCKA offers several advantages over competing metrics: it measures directly on the learned parameters (weights) of the model, reflects weight space symmetries, is independent of probing samples, and is computationally efficient.

In order to provide a first exploration of the degree to which is applicable outside of fully connected neural networks, we investigated its performance in a simplified implementation of transfer learning. We believe that the consistently high performance of wCKA when estimating the similarity of fine-tuned convolutional neural networks highlights its potential. As wCKA is agnostic to feature extraction layers, it can be incorporated into evaluating the fine-tuning process of dense layers with more sophisticated feature extractor architectures, such as Autoencoders Landi et al..

We are *not* yet presenting here a formulation of wCKA that is able to estimate the similarity of fully-tunable architecture, including convolutional layers, layers with residual connections, or batch normalization layers. Nonetheless, we believe that our study already presents compelling evidence for the potential impact of wCKA in helping researchers better characterize and understand what a deep learning model has learned and how that learning changes under different perturbations. For example, our approach could be used to guide the pruning and compressing of models, for building truly diverse ensembles of models, or for ensuring model similarity to a trusted model. We believe that further exploration of how to measure weight similarity for more complex structures is an important future direction that will bear significant fruit.

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

## A  APPENDIX

Figure 5: **Dependency of representational metrics on types of probing samples.** Enhanced box plots show distributions of similarity values of different metrics for random NNs probed by different types of sample corruptions. For Procrustes and CKA, some out-of-distribution corruptions, such as "fog", introduce more spurious similarity than others, such as "impose noise". Similarly, for dCKA, though the mean similarity value centers around zero, some out-of-distribution corruptions, such as "fog", introduce more variance than others, such as "impose noise".

