# OpenReview forum: "Benchmarking a well-calibrated measure of weight similarity of deep neural network models"
_ICLR.cc/2025/Conference — Submitted to ICLR 2025_

### Official Review · Reviewer_VvhP · 2024-10-29

**Soundness:** 2
**Presentation:** 3
**Contribution:** 2
**Rating:** 3
**Confidence:** 4

**Summary:**

This paper addresses the challenge of the lack of well-calibrated metrics for quantifying the similarity between neural networks. Instead of computing similarity between representations, the authors propose wCKA to measure the similarity between weights. wCKA is invariant to neuron permutation, independent of specific input samples, and computationally efficient. wCKA can provide well-calibrated similarity compared with representation similarities, such as CKA and dCKA, and aligns better with functional similarities across different architectures, training methods, and initializations.

**Strengths:**

The method is well-motivated and is interesting to the interpretable AI community. The proposed wCKA focuses on weight similarity, which is a novel approach, and unlike CKA and dCKA, its computation is independent of the number of samples.  wCKA is well-calibrated and correlates well with functional similarities.

**Weaknesses:**

My biggest concern is that the wCKA only applies to neural networks with specific architectures, such as fully connected neural networks, and it is not easy to be generalized for other architectures, such as transformer, which are the mainstream of deep learning.

The evaluation is limited. Although the paper shows promising results, most of the experiments are on MNIST. It would be interesting to see how wCKA performs on diverse real-world datasets, such as larger image datasets (ImageNet or CIFAR-100) and text datasets (GLUE benchmark).

**Questions:**

1. Can wCKA be applied easily to other architectures, such as transformer?

2. Can wCKA be applied to a specific layer, or must it be the whole network? Would applying a layer-wise wCKA affects the conclusion if possible?

3. If wCKA can only be applied to the whole network, which layer did you apply CKA/dCKA on in the benchmarking?

4. Although the computation is independent of the number of samples, it depends on the number of parameters. All models tested in the paper are small. Would wCKA be scalable to large architectures with billions of parameters? It would be great to have any theoretical or empirical analysis of wCKA's computational complexity as the number of parameters increases,

5. It would be great if the author could provide examples of real-world applications where wCKA is more favorable than existing methods due to its unique properties, such as sample-independence and computational efficiency.

**Details Of Ethics Concerns:**

There is no concern about ethics.

---

> ### Author Response · Authors · 2024-11-12
> **What do we need to do?**
>
> Dear Reviewer:
>
> Thank you very much for your careful and thoughtful review of our manuscript. We would like to revise the manuscript in order to address your concerns and try to see if we can take the paper over the very high threshold for publication in ICLR.
>
> When preparing the manuscript, we were concerned by the possibility that the inclusion of too many results would make it too long and too onerous to verify. For that reason, we decided to focus our experimental studies on the simplest cases (both in terms of tasks and models). However, as another reviewer wrote, there is no reason why our approach cannot not be applied to much more complex tasks or models.
>
> In the same spirit as your concerns, another reviewer wrote that 'the method being presented isn’t extended to any type of modern deep learning architectures, i.e. convolutional layers (although this might be possible as discussed in (Wang et al., 2022)), residual connections, attention mechanisms / transformers.'
>
> As we wrote to that reviewer, we believe that at this stage we are not able to rigorously extend our analysis models with attention mechanisms / transformers.
>
> However, we believe that we are able to address the question of convolutional layers and residual connections.
>
> Would be sufficient in order to bring the manuscript over the threshold for publication if we studied:
>
> 1. a more complex set of models that include convolutional layers and residual connections?
> 2. a more complex set of tasks such as:
>          a. CIFAR-10?
>          b. ImageNet?
>
> We eagerly await your answer.
>
> Sincerely,
> -- the authors

---

### Official Review · Reviewer_rEok · 2024-11-04

**Soundness:** 3
**Presentation:** 3
**Contribution:** 2
**Rating:** 5
**Confidence:** 3

**Summary:**

This paper proposes a new way of comparing neural networks. While the vast majority of such methods operate on the hidden activations of the two networks (working on the assumption that two models are similar if they represent the same data in a similar way), this work proposes a similarity function based on the network weights themselves, wCKA. By looking at the weights directly and performing several normalizations, wCKA attempts to overcome a number of issues that plague existing work such as a strong dependence of hyperparameter choices. At a high-level, the method consists of applying the well-known similarity method for activations, centered kernel alignment (CKA), to a normalized version of the network weights. The work then shows that wCKA is invariant to various similarities in architecture that do not result in functional differences in the network. Finally, the paper describes a range of small-scale experiments which suggest that wCKA is well-calibrated and can “see past” some differences in training and architecture to align with functional similarity.

**Strengths:**

**Clarity:** The paper is well-written and clear. It gives a good introduction to a major problem in the field of representation similarity: that the similarity of representations can depend heavily on training hyperparameters and other details and thus do not capture what researchers really want when comparing models. With the exception of a few points noted below, the background section is concise, getting the main ideas across without introducing unnecessary mathematics.

**Problem importance:** The reviewer believes that this is an important and fundamental problem that remains unsolved. While the reviewer does not agree with statements in the introduction and abstract that suggest that this is the major barrier needed to overcome network fragility, it is definitely a challenge that deserves deep thought from the field.

**Approach:** The approach of looking directly at the weights of a network is natural and it is surprising that more works have not yet explored it. By looking at the weights of a network one avoids the additional dependence on choice of activation data. As raw weights are challenging to interpret, the reviewer would have been interested in understanding what structural features of the weights are weighing most heavily in the similarity measurements (even at an informal level).

**Weaknesses:**

Overall, this reviewer thinks that this paper would benefit from more and wider experiments. The two major areas where things could be further fleshed out include:

**More realistic networks in experiments:** Unfortunately, where similarity functions are likely needed the most is in the setting of large, real-world models. For instance, LLMs, ResNet50 sized classifiers, object detectors, and various flavors of generative image models. While it makes sense to start studies from small MLPs, it would have been great to see a proof-of-concept that this works on large architectures trained on large and high-dimensional datasets. This is particularly important because, beyond compute considerations alone, some analytical methods tend to fail as models are scaled-up (potentially because of concentration of measure type phenomena). While there are many models and datasets that would work for this, something like a ResNet50 for vision models and Llama3-8B for language models would be reasonable targets. In terms of datasets, any vision dataset with larger ambient dimension (e.g., ImageNet) would be interesting to see.

**More granular results:** The reviewer finished the work feeling that they still don’t have a sense of the quirks of the wCKA similarity function. All similarity functions are sensitive to some types of model differences over others. Sometimes this follows from basic properties of the comparison method. At other times, particularities only become apparent via fine-grained testing. For instance, heatmaps with individual comparisons of layers would have been useful to get a feel for wCKA. An interesting experiment (which has been applied before for representation similarity functions) is to look at similarity between different layers of a single model. It would be interesting to compare this to the equivalent CKA experiment. It would also have been useful to compare the same architecture trained on different datasets, does wCKA show these as different?

**Limitations:** In line with the problem of getting to know wCKA better, it would be useful to describe its limitations. Either limitations that are intrinsic to its definition or empirically observed limitations. Examples include limitations to the types of layers to which wCKA can be applied and/or compared. I know some of this is directly inherited from CKA, but it would be good to state explicitly as some of this has different implications when weights are used. When we apply CKA to activations arising from the same subset of data, we can compare them even though the feature space dimensions may be different because the number of instances is the same. On the other hand, when one does this for weights, such a comparison is not possible I think?

### Nitpicks:
Line 033: The paper lists LLMs and transformers are distinct examples of recent deep learning architectures. While it is undoubtedly true that LLM differ from small transformers in critical ways because of their scale, it is not clear that this is an example of the “dizzying introduction of novel model architectures”.
Line xxx: “Procrustes measure the similarity…” $\mapsto$ “Procrustes measures the similarity…”
There a several equivalent definitions of Procrustes measure, they have different merits. The one provided in the paper is useful for computation. Another, $\min_{A \in O(d)} ||X_1 - AX_2||_F$ is good for intuition (find the orthogonal transformation that makes $X_1$ look most similar to $X_2$). The reviewer thinks that including the second one might be slightly better in this background section.
Line 166: Should the $\ell$ in $\mathbb{R}^{d_i^{i-1} \times d_i^\ell}$ be $i$?

**Questions:**

- Ultimately, we want similarity metrics to give us greater insight into our models. Has wCKA provided any insights not found with other comparison methods?
- Are there any limitation on the types of layers wCKA can be applied to?

---

> ### Author Response · Authors · 2024-11-12
> **What do we need to do?**
>
> Dear Reviewer:
> Thank you so much for your encouraging comments 'that this is an important and fundamental problem that remains unsolved' and that 'looking directly at the weights of a network is natural and it is surprising that more works have not yet explored it.' Additionally, we want to thank you for your thoughtful comments and questions.
>
> We apologize that our approach focused too much on simplicity and did not extend to more realistic tasks.  We decided (incorrectly) that simplicity would better enable a rigorous evaluation of our claims, but understand that practical applications require testing on more realistic cases.
>
> We would like still to attempt to overcome the very high threshold for publication in ICLR.  With that goal in mind, we respectfully ask whether considering more realistic cases would move the needle in your evaluations.  As another reviewer wrote, it is simple enough to extend our study to more complex tasks and models.
>
> Thus, we ask: would it be sufficient in order to bring the manuscript over the threshold for publications if we studied*:
>
> 1. a more complex tasks such as CIFAR-10?
> 2. a more complex set of CNNs (with fixed backbones) and fully-connected models?
>
> *While you mention LLM in your comments, we do not believe we are at a stage where we could rigorously extend our analysis to such models, so we will not be able to extend this work in this manner at this time. However, we hope that the large number of other DL models to which our approach would apply would be relevant to the field at large.
>
> We eagerly await your answer.
>
> Sincerely,
> -- the authors

---

> > ### Comment · Reviewer_rEok · 2024-11-23
> > **Response to the authors**
> >
> > The reviewer would like to see that the method is useful for models designed for higher dimensional and more challenging datasets, so something like ImageNet (as pointed out by another reviewer, there are now MLPs that can achieve reasonable accuracy on this). It also seems that at a minimum, the method should be shown to work with convolutional networks (even if there isn't currently a way to apply wCKA to transformers).

---

### Official Review · Reviewer_Fm22 · 2024-11-07

**Soundness:** 3
**Presentation:** 3
**Contribution:** 2
**Rating:** 3
**Confidence:** 4

**Summary:**

The paper presents a novel model similarity metric, weights centered kernel alignment (wCKA), which compares the weights of the models directly instead of the internal representations / activations like past works have done. The method is a combination of the weight normalization operator and CKA, both of which have been proposed by past works. The authors compare their proposed method with three popular model (dis)similarity metrics often used in the literature, namely Procrustes distance, CKA and deconfounded CKA. They use a benchmarking framework correlating the various model similarity measures with model functionality. They show through statistical testing that their method is better "calibrated" and better captures the functional similarities between different models than the considered benchmark methods.

**Strengths:**

- I do believe the problem being tackled is an important one and better model similarity measures would greatly benefit the field as a whole.
- The paper is generally well written and clear although some sections would require further clarifications (see the other sections of this review)
- The proposed method is interesting and takes a different approach to model comparison, namely comparing weights, than past methods which have focused on comparing representations. The method is theoretically sound as far as I can tell.
- I like the comparison / benchmarking framework proposed, which follows in the steps of Ding et al., 2021 relating model similarity to model functionality. Furthermore, using the agreed predictions between two models across standard test samples, OOD corruptions and adversarial samples is a notable improvement over simply utilizing the accuracy on a given task. This benchmarking framework, or a similar one, should be used in papers in this field.
- The benchmarked methods (Procrustes, CKA, dCKA) are relevant.
- I appreciate the fact that the authors have run statistical tests to support their conclusions.

**Weaknesses:**

While I do generally like the quality of the paper there are few important weaknesses that prevent me from giving a higher score. These weaknesses are presented below, in order of importance.

- **The scope of the method is inherently limited.** From my understanding, the proposed method relies on the models being MLPs, i.e. simple feedforward linear networks, and the experiments show this, all experiments being done with standard MLPs. Even in section 3.3 where a pre-trained conv net back-bone is used, it’s the MLP heads which are then fine-tuned that are compared (as far as I can tell, see questions). I consider this to be the main limitation of the work. The method being presented isn’t extended to any type of modern deep learning architectures, i.e. convolutional layers (although this might be possible as discussed in (Wang et al., 2022)), residual connections, attention mechanisms / transformers. Since model similarity measures are inherently practical tools, i.e. they are meant to be used in practice to evaluate and compare models, the fact that the proposed method doesn’t extend to *any* modern model architectures, which are used by practitioners and researchers alike, will significantly limit the impact of this work. The authors do acknowledge this weakness in the last paragraph of the discussion, which I appreciate, but it doesn’t do anything to actually remedy this weakness.
- **A more detailed discussion of the implications of switching from representation based model similarity metrics to a weights based similarity metrics is missing.** The switch from representation based model comparison to weights based model comparison is significant and hasn’t been properly discussed anywhere in the paper. For example, in Figure 1 the authors show that their method yields ~0 similarity between randomly initialized neural networks and use this as a justification to say that their method is “better calibrated”. However, this might simply be an artefact of their method being weights-based instead of representations-based. The Johnson-Lindenstrauss lemma, a classical result in data science, states that a set of high dimensional points can be embedded into a space of much lower dimension in such a way that distances between points are approximately preserved through random orthogonal projections. In a sense, multiplication by a randomly initialized weight matrix is a random projection, therefore I expect even randomly initialized neural network to have some similarity in their representations. Therefore the fact that the other, representation based, similarity measures don’t have a score of ~0 when comparing randomly initialized network isn’t necessarily a “bug” as the authors make it out to be.
- The scope of the empirical evaluations is limited. The paper presents a novel similarity metric but does little to justify the method's usefulness. In other words, what characteristic of neural networks can be observed using this method that couldn't be observed with past methods?
- The novelty and contribution of this work is somewhat limited since wCKA combines the normalization operator from (Wang et al., 2022) with CKA (Kornblith et al., 2019). A more detailed comparison with (Wang et al., 2022) would be appreciated, this is however relatively minimal.

Based on the identified weaknesses, especially the first one, I currently do not recommend this work for publication at ICLR. I encourage the authors to either publish the work with minor modifications at smaller venues or significantly improve the method to be able to apply it to more modern architectures and try again to submit to a large scale conference. I’m open to hearing the authors’ point of view on these weaknesses and changing my score if adequate justification is provided but unless I severely misunderstood something critical about the paper I don't foresee this happening.

**Questions:**

Here I list my questions as well as more minor weaknesses and comments which should be addressed in future iterations of this work but are unlikely to influence my score.

- In the abstract the authors write “… similarity of the internal representations of models…” while their method isn’t concerned with representations, I recommend writing "similarity of models" directly like is done in the text.
- First paragraph of the intro: “and, most recently, large language models.” LLMs are not a exactly a novel “model architecture”, I would remove it from the list.
- The introduction section is somewhat lengthy and convoluted. While I agree with the important “knowledge gap at the heart of the field” I don't think a single model similarity measure, no matter how advanced, is likely to fully bridge that gap.
- The methods section is somewhat lacking in terms of explanations and intuition.
    - The description of the Procrustes method, i.e. “minimizing the Frobenius norm of the difference between the two matrices”, isn’t quite accurate and could be more detailed while still remaining brief.
    - Eq. 2 is the empirical estimator of HSIC. This is a detail but is worth mentioning.
    - Section 2.2 where wCKA is introduced would benefit from a better description of the weight normalization operator (on which wCKA is based) and of the wCKA method itself. Right now the section is mostly comprised of equations with little or no explanations or intuition provided as to what the normalization or the method are doing.
- The authors use $xW$ to denote the multiplication of a linear layer’s weights with its inputs. This is a bit arbitrary but I believe the $Wx+b$ formulation is more popular and will be more familiar to readers so I would suggest using that formulation instead.
- Multiple references are not well cited, for example Klabunde et al., Landi et al. and Ferreira et al. are lacking the year of the publication.
- In section 2.4 the “S” used in the equation is not properly defined.
- While I generally like the benchmarking framework described in section 2.4, the proposed wCKA method does not require any data therefore none of the “clean test samples, OOD corruptions, and adversarial attacks” will affect wCKA. A proper discussion of this would benefit the paper.
- The concept of “calibration” is used throughout the paper, even in the title, but there is no clear definition of what a “well-calibrated” method signifies anywhere in the text. While I would agree that the concept of "calibration" is hard to define, it is still important to provide a definition of the term, even if it's just in the context of this specific work.
- Section 3.3 is misleading since it gives the impression that the proposed method can be applied to convolutional neural networks but from my understanding wCKA is only applied to the fine-tuned MLPs which are added on top of the pre-trained convolutional backbone (unless I misunderstood?). Either way this needs to be made clearer in the text.

---

> ### Author Response · Authors · 2024-11-12
> **What do we need to do?**
>
> Dear Reviewer:
>
> Thank you very much for your careful and thoughtful review of our manuscript. We would like to revise the manuscript in order to address your concerns and try to see if we can take the paper over the very high threshold for publication in ICLR.
>
> When preparing the manuscript, we were concerned by the possibility that the inclusion of too many results would make it too long and too onerous to verify. For that reason, we decided to focus our experimental studies on simplest cases (both in terms of tasks and models). However, as another reviewer wrote, there is not reason why our approach cannot not be applied to much more complex tasks or models.
>
> In your report, your write that 'the method being presented isn’t extended to any type of modern deep learning architectures, i.e. convolutional layers (although this might be possible as discussed in (Wang et al., 2022)), residual connections, attention mechanisms / transformers.'
>
> We believe that at this stage we are not able to rigorously extend our analysis models with attention mechanisms / transformers.
>
> However, we believe that are able to address the question of convolutional layers and residual connections.
>
> Would be sufficient in order to bring the manuscript over the threshold for publication if we studied:
>
> 1. a more complex set of models that include convolutional layers and residual connections?
>
> We eagerly await your answer.
>
> Sincerely,
> -- the authors

---

> > ### Comment · Reviewer_Fm22 · 2024-11-14
> >
> > I would be willing to change my score to a 5 if a convincing analysis is provided regarding how this method can be applied to convolutional layers and residual connections, as well as accompanying experiments. To clarify, I would need to see how this method can be applied *to the convolutional layers and residual connections themselves* and not just to fully connected linear layers which are part of a network also containing convolutional / residual connections. My last question still remains, for the results in section 3.3 it is unclear if the method is applied to the convolutional layers themselves or only to the fine-tuned linear layers, this needs to be clarified.
> >
> > The authors would also need to address my other concerns.

---

> > > ### Author Response · Authors · 2024-11-14
> > > **Thank your for your feedback**
> > >
> > > Dear reviewer Fm22,
> > >
> > > Thank you for your timely response.
> > >
> > > At this stage, the results in Section 3.3 apply only to fine-tuned linear layers. We are actively working to extend our method to convolutional layers themselves and residual connections, as you suggested. We will address this and your other concerns in the revised manuscript.
> > >
> > > Sincerely,
> > >
> > > The Authors

---

### Official Review · Reviewer_ZiAn · 2024-11-08

**Soundness:** 2
**Presentation:** 2
**Contribution:** 2
**Rating:** 3
**Confidence:** 4

**Summary:**

Examines the correlation between weight similarity measures [1] and the functional similarity of models, showing that in many cases such measures perform better than representation similarity. They improve on the set-up of [2] by using the fraction of agreed predictions instead of the (coarser) accuracy gap. The paper also examines ’intuitive’ calibration tests that compare models with random weights and trained models. The weight similarity metric, wCKA, is motivated by showing that it is invariant to intertwiner groups, groups that transform model weights such that the underlying network function is unchanged but the representations are different, a desirable property for a measure on model weights.

**Strengths:**

- If it holds up, the core result— that comparing just the weights of models often performs better on statistical testing for functional similarity than representation similarity— is both interesting and surprising. The method also is substantially more efficient, does not require data, and may be complementary to representation similarity methods.
- The experiments are a decent start to begin to validate the above claims.
- The theoretical grounding is nice, and supports the experimental claims made in the paper. In particular, connecting chain normalization [1] (which seems to me to be an under-cited paper!), showing invariance to intertwiner groups [3], and using CKA [4].

**Weaknesses:**

- The current experiments are not nearly sufficient to justify the usefulness of the metric. The models considered are far too small (all less than 2500 neurons) and trained on too toy of problems (MNIST). The scope and size of the model architecture and datasets examined (e.g., there are large fully connected models that achieve good enough performance on ImageNet or CIFAR-10 or text tasks) should be increased.
- There is a large related literature on learning on learning on models (eg [5]) that the authors do not cite / are not aware of and should probably be used to contextualize (or strengthen) these results.


[1] Wang, Guangcong, et al. "Understanding weight similarity of neural networks via chain normalization rule and hypothesis-training-testing." arXiv preprint arXiv:2208.04369 (2022).

[2] Ding, Frances, Jean-Stanislas Denain, and Jacob Steinhardt. "Grounding representation similarity through statistical testing." Advances in Neural Information Processing Systems 34 (2021): 1556-1568.

[3] Godfrey, Charles, et al. "On the symmetries of deep learning models and their internal representations." Advances in Neural Information Processing Systems 35 (2022): 11893-11905.

[4] Kornblith, Simon, et al. "Similarity of neural network representations revisited." International conference on machine learning. PMLR, 2019.

[5] Lim, Derek, et al. "Graph metanetworks for processing diverse neural architectures." arXiv preprint arXiv:2312.04501 (2023).

**Questions:**

- Why consider only extremely small architectures and very small datasets? The method seems very cheap to use.
- Why not extend to the same set of architectures considered in [1]?
- Could the authors better motivate the transfer experiments in Fig. 4? They seem prima-facie designed to adversely select against representation dissimilarity metrics?

---

> ### Author Response · Authors · 2024-11-12
> **What do we need to do?**
>
> Dear Reviewer:
> Thank you very much for your careful and thoughtful review of our manuscript.  We would like to revise the manuscript in order to address your concerns and try to see if we can take the paper over the very high threshold for publication in ICLR.
>
> When preparing the manuscript, we were concerned by the possibility that the inclusion of too many results would make it too long and too onerous to verify. For that reason, we decided to focus our experimental studies on simplest cases (both in terms of tasks and models). However, as you write, there is not reason why our approach cannot not be applied to much more complex tasks or models.
>
> Would be sufficient in order to bring the manuscript over the threshold for publications if we studied:
>
> 1) a more complex tasks such as  CIFAR-10?
> 2) a more complex set of CNNs (with fixed backbones) and fully-connected models?
>
> We eagerly await your answer.
>
> Sincerely,
> -- the authors
>
> PS: Could you clarify for us what you meant by the statement: 'They seem prima-facie designed to adversely select against representation dissimilarity metrics?'

---

> > ### Comment · Reviewer_ZiAn · 2024-11-13
> > **Some experiments you could do**
> >
> > Thanks, looking forward to seeing your method validated in more convincing settings. Here are two additional groups of experiments that, taken together, I would find convincing:
> >
> > - I strongly encourage you to scale your experiments to ImageNet (and also CIFAR100/CIFAR10). Eg you might find the set-up and recipe (and checkpoints) in [1] useful (and plausibly [2] if you wanted to express MLP-mixers as vanilla fully connected models for your algorithm. I also think a similar trick [3] should be doable for expressing CNN weights as Toeplitz matrices?).
> >
> > - So you can apply this method to more compelling architectures and tasks, I also encourage you to test some less principled variants of chain normalizing weight matrices. In particular, I would find it compelling if you provided such results for transformer models on text tasks (perhaps using a reasonable but not principled approach [eg not permutation invariant] to handling attention layers and residual connections).
> >
> > Regaring my point about adversely selecting against rep. dis. metrics, I think reviewer Fm22 better expresses this point in the bullet titled '**A more detailed discussion of the implications of [...]**'
> >
> >
> > [1] Bachmann, Gregor, Sotiris Anagnostidis, and Thomas Hofmann. "Scaling mlps: A tale of inductive bias." Advances in Neural Information Processing Systems 36 (2024).
> >
> > [2] Hayase, Tomohiro, and Ryo Karakida. "Understanding MLP-Mixer as a wide and sparse MLP." arXiv preprint arXiv:2306.01470 (2023).
> >
> > [3] Sedghi, Hanie, Vineet Gupta, and Philip M. Long. "The singular values of convolutional layers." arXiv preprint arXiv:1805.10408 (2018).

---

> > > ### Author Response · Authors · 2024-11-13
> > > **Thank you for constructive advice!**
> > >
> > > Dear Reviewer ZiAn,
> > >
> > > Thank you very much for your constructive advice and for highlighting valuable resources and promising directions!
> > >
> > > We are currently in the process of implementing our algorithm with scaled MLPs as described in [1] and are enthusiastic about the possibility of extending our method to convolutional layers using the Toeplitz matrix trick. We will keep you updated as we obtain further results.
> > >
> > > Once again, we truly appreciate your insightful feedback and the opportunity to strengthen our manuscript.
> > >
> > > Sincerely,
> > >
> > > The Authors

---

### Author Response · Authors · 2024-11-12
**Thank you for encouraging comments**

Dear reviewers:

We want to thank you all for your very careful and thoughtful reviews.

We were very encouraged by the numerous and significant strengths that you all identified in our study. Namely:

1. That we address an important but unsolved problem and provides a good introduction to a major problem in the field.
2. That our study is well written and clear, it is well-motivated, and that better model similarity measures would greatly benefit the field.
3. That comparing the weights of models:
             (i). is an interesting original approach that approaches model comparison from a different perspective.
             (ii) is independent of the number of samples.
             (iii) is well calibrated measure.
             (iv) performs better in a statistical sense for measuring functional similarity than representation similarity.

4. That we offer a nice theoretical grounding that is sound and supports the experimental results.
5. That we use a rigorous statistical approach in our comparisons

We are grateful that you consistently identified some steps that we could take to address the weaknesses of the study.

Thus, we are hoping to be given the chance of addressing those weaknesses in a revision that would occur during the next ten days.

We eagerly await your answers.

Sincerely,
--the authors

---

### Author Response · Authors · 2024-11-22

Dear Reviewers,

Thank you again for your thoughtful comments and constructive feedbacks.

When applying wCKA in its current form to scaled MLPs with residual connections, we observe that the correlation between our metric wCKA and functional similarity weakens, to an extent that does not outperform existing metric like deconfounded CKA. We suspect this is because residual connections reduce the number of intertwiner transformations, and wCKA potentially biases towards higher similarity scores.

When extending wCKA to convolutional layers using the reshaping trick for convolutional kernels as in Wang’s paper (which introduced the chain normalization operator), our preliminary results also show a weakened correlation with functional similarity. On the other hand, using the Toeplitz matrix trick would result in very large, sparse matrices, significantly impacting computational efficiency.

Given these challenges, we are unable to provide convincing results for residual connections and convolutional layers within the limited time available. That said, we believe that the tunable weight matrices reflect intrinsic "knowledge" learned by the model, and with further adjustment, wCKA will serve as a unique window to look into large, opaque models.

Once again, thank you for your valuable time and feedback. We would especially like to thank Reviewer ZiAn for helpful resources. If you have any additional suggestions, resources, or advice, we would greatly appreciate them. We look forward to sharing an improved version of wCKA in potential future submissions.

Sincerely,

The Authors

---

### Meta-Review · Area_Chair_Hxtf · 2024-12-19

**Metareview:**

The paper investigates a new weight-based similarity metric, wCKA, designed to measure how closely two neural networks align in terms of their learned weight structures, rather than their representations. Empirically, wCKA is shown to correlate well with functional similarity (i.e., models making similar predictions), especially in simple settings (e.g., MLPs on MNIST). It outperforms common metrics like representation-based CKA and deconfounded CKA in terms of calibration, and it avoids the confounders tied to input data distribution. Strengths include the paper’s conceptual clarity, theoretical grounding, and the demonstration of wCKA’s invariances. Weaknesses stem largely from the limited scope of experiments—only small MLPs and simple tasks (like MNIST) are considered, leaving questions about wCKA’s applicability and scalability to modern, large-scale architectures (e.g., CNNs, transformers) and more complex datasets (e.g., CIFAR-10, ImageNet). The work’s current form lacks broad empirical validation, thus the authors are encouraged to revise and resubmit.

**Additional Comments On Reviewer Discussion:**

During the rebuttal period, reviewers repeatedly requested evidence that wCKA can scale to more realistic deep learning scenarios, including CNNs, residual connections, and more challenging datasets like CIFAR-10 or ImageNet. The authors responded by promising to investigate convolutional architectures and scaled MLPs, and they acknowledged the difficulty in extending to transformers in the short term. No additional experiments were provided before the rebuttal deadline, leading to a consensus towards rejection.

---

### Decision · Program_Chairs · 2025-01-22

Reject